# The Australian Injury Comorbidity Indices (AICIs) to predict in-hospital complications: A population-based data linkage study

Dasamal Tharanga Fernando[1]*, Janneke Berecki-Gisolf[1], Stuart Newstead[1], Zahid Ansari[2]

1 Monash University Accident Research Centre, Monash University, Clayton Campus, Clayton, Victoria, Australia, 2 Victorian Agency for Health Information, Melbourne, Victoria, Australia

* tara.fernando@monash.edu

## Abstract

### Background

Hospital-admitted patients are at risk of experiencing certain adverse outcomes during their hospital-stay. Patients may need to be admitted to the intensive care unit or be placed on the ventilator while there is also a possibility for complications to develop. Pre-existing comorbidity could increase the risk of these outcomes. The Charlson Comorbidity Index (CCI) and the Elixhauser Comorbidity Measure (ECM), originally derived for mortality outcomes among general medical populations, are widely used for assessing these in-hospital complications even among specific injury populations. This study derived indices to specifically capture the effect of comorbidity on intensive care unit and ventilator use as well as hospital-acquired complications for injury patients.

### Methods

Retrospective data on injury hospital-admissions from July 2012 to June 2014 (161,334 patients) for the state of Victoria, Australia was analysed. Results from multivariable regression analysis were used to derive the Australian Injury Comorbidity Indices (AICIs) for intensive care unit and ventilator hours and hospital-acquired complications. The AICIs, CCI and ECM were validated on data from Victoria and two other Australian states.

### Results

Five comorbidities were significantly associated with intensive care unit hours, two with ventilator hours and fifteen with hospital-acquired complications for hospitalised injury patients. Not all diseases listed in the CCI or ECM were found to be associated with these outcomes. The AICIs performed equally well in terms of predictive ability to the long-listed ECM and in most instances outperformed the CCI.

**Data Availability Statement:** Data cannot be shared publicly because of confidentiality clauses. Data are available from the CVDL, CHeReL and WADLS Institutional Data Access / Ethics

Committee for researchers who meet the criteria for access to confidential data. Data access queries may be directed to Monash University Human Research Ethics Committee (muhrec@monash. edu) or the data custodian (cvdl@dhhs.vic.gov.au), to the New South Wales Population and Health Services Research Ethics Committee (CINSW-Ethics@health.nsw.gov.au), and to the Department of Health WA Human Research Ethics Committee (HREC@health.wa.gov.au).

**Funding:** This paper is part of a PhD thesis which is supported by the Victorian Injury Surveillance Unit (VISU) funded by the Victorian Government. The PhD student received the Australian Government Research Training Program scholarship stipend during the course of the project. This PhD project was initiated by VISU and therefore funds for purchasing data were supplied VISU. TF and JB are staff of VISU; however, the role of VISU was solely to be a source of data acquisition funding and supervision support. The funders had no role in study design, data collection and analysis, decision to publish, or preparation of the manuscript.

**Competing interests:** The authors have read the journal's policy and have the following potential competing interests: TF has received funding support from the Victorian Injury Surveillance Unit (VISU) at Monash University to pay for two datasets, and received PhD supervision. This does not alter our adherence to PLOS ONE policies on sharing data and materials. There are no patents, products in development or marketed products associated with this research to declare. The other authors have declared that no competing interests exist.

## Conclusions

Associations between outcomes and comorbidities vary based on the type of outcome measure. The new comorbidity indices developed in this study provide a relevant, parsimonious and up-to-date method to capture the effect of comorbidity on in-hospital complications among admitted injury patients and is better suited for use in that context compared to the CCI and ECM.

## 1. Background

Hospital-admitted patients can face certain adverse outcomes during their hospital stay. These include: admission to the intensive care unit (ICU), being placed on the mechanical ventilator (MV), development of complications and even mortality. ICU stay, MV use and complications result in increased burden (such as cost and length of stay (LOS)), and complications can also lead to mortality [1–3]. Among hospital-admitted patients in Australia, the proportion of patients with at least one hospital-acquired complication was around 6.7% in 2010–11 [4]. Complications have been found to increase the risk of in-hospital death (7 times the risk) and to increase the length of stay (four times the mean LOS) while adding to the cost [4, 5] compared to those without complications.

A recent study found that patients admitted to hospital in relation to injury are more likely to develop complications than general admissions (Moor et al., 2015) [6]. It found that hospital-acquired complications occurred in 13.9% of trauma patients, which was three times the proportion for general admissions. Other studies have reported various proportions of hospital-acquired complications among injury patients: 36.6% by Hoyts et al. (2003) [7], 10% by Holbrook et al. (2001) [8] and 13.3% by Fernando et al. (2019) [9]. As has been reported for general patients, complications increase costs [3], LOS and mortality [2, 6] for injury patients.

Another contributor to adverse outcomes are pre-existing comorbidities. Ahmad et al. (2007) [10] showed that patients with diabetes mellitus were 1.8 times more likely to develop complications than those without, and they had a 50% greater ICU-LOS compared to those without complications. Among seriously injured older adults, pre-existing comorbidity increased complications by three-fold [11]. Senn-Reeves et al. (2015) [12] also showed that comorbidity and injury characteristics were associated with injury-related complications.

The methods used to capture comorbidity in the past to assess their associations with outcomes are numerous. Most studies used a study-specific list of comorbidities [11, 13–15] or an abbreviated list from the National Trauma Data Bank [12]. Others used the Charlson Comorbidity Index (CCI) [16–18], a weighted index derived to predict mortality outcomes. The CCI is rather dated since its initial derivation was in 1987 and significant medical advances that have taken place since then which might impact the relationship between comorbidities and outcomes. The Elixhauser Comorbidity Measure (ECM) is another measure used in epidemiological research (though less often than the CCI) for capturing comorbidity [19]. The Mortality Risk Score for Trauma (MoRT) [20] derived for serious injury patients is the only injury-specific comorbidity index available at present. It was derived using a serious injury cohort so would be less versatile, given its specificity, in being generalised for all injury patients. The evidence also suggests that the ability of a comorbidity indicator to assess risk depends on the outcome being studied [19, 21, 22]. Furthermore, the type of data available and prevalence of specific comorbidities in the population of interest is likely to be relevant when creating an indicator for comorbidity.

## 1.1 Study aims

The purpose of this study is: (1) to develop and validate new indices to assess the impact of comorbidity on outcomes of ICU stay, MV use and hospital-acquired complications using Australian administrative datasets and (2) to compare the performance of the new indices with the CCI and ECM.

## 2 Methods

### 2.1 Data sources

An observational study of existing hospital morbidity data was carried out. Retrospective Australian morbidity data for hospital-admitted injury patients were sourced from the Victorian Admitted Episodes Dataset (VAED), the Admitted Patient Data Collection (APDC) and the Hospital Morbidity Data Collection (HMDC), provided by the Centre for Victorian Data Linkage (CVDL) in Victoria, the Centre for Health Record Linkage (CHeReL) in New South Wales (NSW) and the Data Linkage Branch (DLB) in Western Australia (WA), respectively. All three datasets contain records of public and private hospital admissions with patient demographics and morbidity information. The morbidity data includes forty diagnosis codes for the VAED, fifty-one for the APDC and seventy-eight for the HMDC, containing disease, injury and external cause variables coded to the International Classification of Diseases Tenth Revision, Australian Modifications (ICD-10-AM) [23].

### 2.2 Data linkage

Records within each morbidity dataset were linked by the relevant data linkage units: (i) using deterministic data linkage for the VAED; (ii) probabilistic matching techniques for the APDC; and (iii) a multi-faceted probabilistic linkage that includes numerous automated and manual sub-processes for the HMDC. Using specific identifiers (unique to each data linkage unit), the records within each morbidity dataset were internally linked to allow for follow-up of hospital admissions subsequent to their index admission record over a period of two years. CVDL estimates the false positive match rate to be between 0.5% to 1%, and the false negative match rate to be between 1–2%. The two rates for the APDC are estimated to be around 0.5% [24]. It is expected that the false negatives in the Western Australian Data Linkage System (WADLS) exceed the number of false positives; the derivation of specific estimates though are not attempted.

### 2.3 Case selection

Any record containing an ICD-10-AM diagnosis code in the range "S00" to "T75" or "T79" in the first appearing diagnosis field in the morbidity datasets was considered an injury case; similar to other national reporting [25]. Cases were selected if they were index injuries (i.e., the first injury record for a patient during the study period) and limited to residents of the relevant state. Changes in care type within the same hospital or inward transfers from other hospitals were considered to be part of one episode if they appeared consecutively. Children below the age of 15 years were excluded when deriving indices as they differ to the rest of the cohort in terms of comorbidity prevalence. The Victorian cohort of adult patients consisted of 161,334 patients with an index injury admission between 01 July 2012 and 30 June 2014, the NSW cohort for the same period consisted of 233,521 patients and WA 84,877 patients.

**2.3.1 Coding of outcomes, factors and comorbidities.** *2.3.1.1 Outcomes.* Three outcomes related to in-hospital complications were coded and modelled for index derivation purposes: hours in the ICU, hours spent on the MV and the number of hospital-acquired complications.

The hospital acquired complications were coded according to the classification of hospital acquired diagnoses (CHADx) [26]. This is a common tool used in hospitals in a number of Australian states. The CHADx is grouped into 17 major classes expanding to 144 subclasses. Complications were determined for all index admissions and related admissions (i.e., complication codes recorded in transfers and statistical separations records, as well as readmissions with the same principal diagnosis code as the index principal diagnosis code or a principal readmission diagnosis code of T79, T80-T89 or T90-T98). Readmissions that took place more than six months after the index admission discharge were excluded. All complications were coded using the CHADx hierarchy and summed to determine the total number of complications.

*2.3.1.2 Explanatory variables (socio-demographics).* The baseline explanatory variables (factors) were age, gender, body-region of injury, injury type, injury severity, SEIFA (Socio Economic Indexes For Areas), country of birth, and geographic region (metropolitan or rural). Injury severity was calculated using the ICD-based Injury Severity Score (ICISS) [27]. SEIFA was classified using the Index of Relative Socio-Economic Advantage and Disadvantage (IRSAD) expressed as state-deciles [28].

*2.3.1.3 Explanatory variable (comorbidity).* The main predictor variable of interest was comorbidity. Comorbidities listed in the CCI [18] and ECM [19] were used in this study, based on the codes supplied in Quan et al. (2005) [29] and Sundararajan et al. (2005) [30]. Thirty-one comorbidity groups were selected for this study. The ICD-10 codes corresponding with these comorbidity groups were searched for in the diagnosis fields of the morbidity datasets with the aid of the condition onset flags. The condition onset flag helps distinguish *comorbidities* from *primary conditions* and *complications*.

## 2.4 Statistical analysis

Associations between comorbidities and the three outcomes were assessed using multivariable regression analysis. The factor variables excluding comorbidity were entered in the baseline models. ICU and MV hours were modelled using linear regression with a log transformation while the number of complications was modelled using negative binomial regression. Adjusted and McFadden's $R^2$ were used to evaluate predictive powers of the models. The baseline models were then modified by adding comorbidity using various techniques: the presence of at least one of the thirty-one comorbidities, the count of comorbidities, a binary representation of each comorbidity, the CCI [18], the updated CCI per Quan et al. (2011) [31], and ECM[19]. The Akaike Information Criterion (AIC) [32] for model fit was used to compare the models.

The binary comorbidity indices were derived using a backward elimination process on the models with all thirty-one conditions fitted as binary variables. The resulting models excluded comorbidities that no longer improved model fit. This was ascertained using the AIC statistic [32]; a difference < 10 between two AICs indicates that the model with the additional factors provides no further improvement to the model fit. Using the reduced binary model, weights were computed for each comorbid condition using the exponents of the parameters for comorbidities. A condition was dropped from the weighted index if the weight was less than 1.2 reflecting the lack of impact of this condition on the outcome, even if statistically significant. For weights above this, scoring was based on the range in which the weight fell; $1.2 \leq$ weight $<1.5$ resulted in a score of 1; $1.5 \leq$ weight $<2.5 = 2$ and so on. The sum of these weights created the summed weighted score, which became the weighted injury comorbidity index.

Five models were compared for best predictive ability. Baseline models included sociodemographic and injury factors; subsequently, comorbidity was introduced as follows: 1) binary

representation, 2) weighted summed score, 3) CCI, 4) Updated CCI and 5) ECM [31]. This process was carried out for each outcome.

Finally, the indices were internally validated in sub-groups (in terms of demographics, injury type and severity) and externally validated in NSW and WA data using the same baseline models and comorbidity indicators. The measures of validation were once again the $R^2$s as performed for the main analysis. Since the $R^2$s are proportions, tests for proportions were carried out to ascertain if the validation $R^2$s were significantly different to the $R^2$ in the main analysis. SAS software, Version 9.4 [33] and Stata 14.0 (StataCorp) [34] was used to analyse the data.

### 2.5 Ethics approval and consent to participate

The study was approved by the Monash University Human Research Ethics Committee (Project no: 1256), the New South Wales Population and Health Services Research Ethics Committee (REF: 2017/HRE0601) and the Department of Health WA Human Research Ethics Committee (RGS0000000613). Historical administrative data was used. The research is low risk in that there was no discomfort or risk of harm to the participants. Name, date of birth and other identifiers were removed from the dataset by the data custodians prior to release of the data to the researchers. Due to the magnitude of the dataset, it was impractical to obtain consent.

## 3 Results

### 3.1 Overview of study population

One-third of the Victorian cohort were older adults ($> = 65$ years of age) and more than half were male (Tables 1 and 2). Thirteen -percent were severely injured (per the ICISS) and nearly sixty-percent of the injuries were to the extremities, while the most common injury type was fracture (41%). Around 3% of patients required an ICU stay and 1.6% were on the mechanical ventilator. For adults, the mean ICU hours was 85.4 (95% CI 82.0 to 88.8) and MV hours was 75.2 (95% CI 70.3 to 80.2). Around 16% had at least one hospital-acquired complication. More than half of those requiring an ICU stay or MV use, and around 42% of those with complications, had at least one comorbidity.

Adult patients with HIV/AIDS, cerebrovascular disease, coagulopathy, obesity and peripheral vascular disease spent five days (120 hours) or more on average in the ICU and on the MV (Table 3). The mean number of complications among adults ranged from 2.9 (95% CI 2.7 to 3.1) to 4.8 (95% CI 4.3 to 5.3) among the thirty-one comorbidities, with the highest mean being for patients with valvular disease (Table 3).

The proportion with complications was highest among patients with cardiac arrhythmias, diabetes without complications, uncomplicated hypertension and renal disease (Table 4). Gastrointestinal, cardiovascular, metabolic disorders and genitourinary complications were the most common types, each accounting for more than 10% of all complications (not shown in Tables).

### 3.2 Multivariable regression modelling

The baseline models (models *i*) differed for each outcome; details are presented in Table 5, with a step by step breakdown presented in S1 Table (Supplemental Digital Content SDC1). The $R^2$ values for the baseline models for ICU and MV hours and complications were 10.4%, 14.5% and 2.9% respectively. Residual plots for ICU and MV hours and predicted vs observed plots for complications are presented in S1 Appendix (SDC1). Interaction effects between age

**Table 1. Characteristics of the Victorian study populations.**

| | Patients admitted[1], n (%) | | | |
|---|---|---|---|---|
| | July 2012 to June 2014, Victoria | | | |
| | n | % | At least one comorbidity (%) | Count of comorbidities, mean (95% CI) |
| Total patients | 161334 | | | |
| **Age group (years)** | | | | |
| 0–14 years | 21240 | 13.2 | 1.4 | 0.01 (0.01 to 0.02) |
| 15–24 years | 23213 | 14.4 | 9.9 | 0.12 (0.11 to 0.12) |
| 25–44 years | 36262 | 22.5 | 13.0 | 0.16 (0.16 to 0.17) |
| 45–64 years | 30799 | 19.1 | 19.3 | 0.25 (0.25 to 0.26) |
| 65–84 years | 31390 | 19.5 | 35.9 | 0.54 (0.53 to 0.54) |
| 85 and over | 18430 | 11.4 | 40.0 | 0.62 (0.60 to 0.63) |
| **Gender** | | | | |
| Male[2] | 89144 | 55.3 | 16.8 | 0.24 (0.23 to 0.24) |
| Female | 72190 | 44.7 | 23.5 | 0.33 (0.33 to 0.34) |
| **Injury severity[3]** | | | | |
| Serious injury (ICISS[4]<0.941) | 20884 | 12.9 | 40.6 | 0.64 (0.63 to 0.65) |
| Other injury (ICISS> = 0.941) | 140450 | 87.1 | 16.7 | 0.22 (0.22 to 0.23) |
| **Grouped body region** | | | | |
| Head/face/neck | 32052 | 19.9 | 19.2 | 0.26 (0.25 to 0.27) |
| Trunk | 20730 | 12.8 | 25.0 | 0.36 (0.35 to 0.37) |
| Upper extremity | 54549 | 33.8 | 9.8 | 0.12 (0.12 to 0.13) |
| Lower extremity | 41528 | 25.7 | 22.8 | 0.34 (0.33 to 0.35) |
| Multiple body regions | 53 | 0.0 | 18.9 | 0.25 (0.09 to 0.40) |
| Unspecified body region | 585 | 0.4 | 22.6 | 0.31 (0.26 to 0.37) |
| Body region not relevant | 11837 | 7.3 | 47.7 | 0.66 (0.65 to 0.68) |
| **Grouped injury type (first occurring)** | | | | |
| Superficial injury | 8055 | 5.0 | 23.8 | 0.34 (0.32 to 0.35) |
| Open wound | 22398 | 13.9 | 14.8 | 0.19 (0.18 to 0.20) |
| Fracture | 66686 | 41.3 | 19.6 | 0.29 (0.28 to 0.29) |
| Dislocation, sprain & strain | 10989 | 6.8 | 8.2 | 0.10 (0.10 to 0.11) |
| Injury to nerves & spinal cord | 2038 | 1.3 | 10.4 | 0.13 (0.11 to 0.14) |
| Injury to blood vessels | 1347 | 0.8 | 11.5 | 0.15 (0.12 to 0.17) |
| Injury to muscle & tendon | 8451 | 5.2 | 9.1 | 0.11 (0.10 to 0.12) |
| Crushing injury | 335 | 0.2 | 3.3 | 0.04 (0.01 to 0.06) |
| Traumatic amputation | 1612 | 1.0 | 7.3 | 0.08 (0.07 to 0.10) |
| Eye injury- excluding foreign body | 512 | 0.3 | 16.4 | 0.22 (0.17 to 0.27) |
| Intracranial injury | 6416 | 4.0 | 29.4 | 0.43 (0.41 to 0.45) |
| Injury to internal organs | 1762 | 1.1 | 23.5 | 0.32 (0.29 to 0.35) |
| Foreign body | 2733 | 1.7 | 11.3 | 0.15 (0.13 to 0.16) |
| Burns | 1879 | 1.2 | 15.1 | 0.21 (0.18 to 0.23) |
| Other and unspecified injury | 14284 | 8.9 | 19.8 | 0.27 (0.26 to 0.28) |
| Systemic-poisoning/toxic effects | 10536 | 6.5 | 49.5 | 0.68 (0.67 to 0.70) |
| Other effects of external cause/complication | 1301 | 0.8 | 32.5 | 0.54 (0.48 to 0.59) |
| **Geographic region** | | | | |
| Metropolitan Area | 118959 | 73.7 | 19.8 | 0.28 (0.27 to 0.28) |
| Rural Area | 42375 | 26.3 | 19.8 | 0.28 (0.27 to 0.29) |
| Unknown | | | | |
| **Outcomes** | | | | |

(*Continued*)

**Table 1.** (Continued)

| | | | Patients admitted[1], n (%) | |
| --- | --- | --- | --- | --- |
| | | | July 2012 to June 2014, Victoria | |
| | n | % | At least one comorbidity (%) | Count of comorbidities, mean (95% CI) |
| Patients requiring an ICU stay | 5285 | 3.3 | 54.3 | 0.89 (0.86 to 0.92) |
| Mean ICU[5]-LOS[6,7] | | | 85.37 (82.00 to 88.75) | |
| Patients requiring MV use | 2495 | 1.6 | 54.7 | 0.85 (0.81 to 0.89) |
| Mean MV[8]-LOS[9] | | | 75.23 (70.27 to 80.19) | |
| Patients with at least one complication[10] | 26127 | 16.2 | 42.3 | 0.68 (0.67 to 0.70) |
| Complications (median, IQR)[11] | 2 (1 to 4) | | 2(1 to 5) | 0.70 (0.69 to 0.71) |
| Complications (mean, 95% CI)[11] | 3.03 (2.99 to 3.06) | | 3.51 (3.45 to 3.57) | |

and sex, age and comorbidities, and sex and comorbidities were also modelled, none of which improved the baseline models' predictive abilities significantly. Therefore, the interaction terms were excluded from further analysis although the margin plots for the interaction terms show some associations between certain comorbidities and the interaction between age and sex (S2 Appendix) (SDC2).

The baseline models with the addition of various existing comorbidity indices are also presented in Table 5 and S1 Table (SDC3.1). Among them are the newly derived binary (model *ii*) and weighted comorbidity indices (model *iii*) and existing indices (CCI (model *iv*), updated CCI (model *v*) and ECM (model *vi*)).

**3.2.1 ICU hours and MV hours.** Assessing model fit using the AICs, the best was model *vi* (containing the ECM), followed by models *iii* (containing the new weighted injury comorbidity index) and *ii* (containing the Australian Injury Comorbidity Index for ICU hours (AICI-icu) with five comorbidities and the Australian Injury Comorbidity Index for MV hours (AICI-mv) with two comorbidities) (Table 5 and S1 Table (SDC3.1)). The CCI (model *iv*) had a poorer fit.

**3.2.2 Hospital-acquired complications.** The best in terms of model fit was once again model *vi* (containing the ECM), followed by model *x* (containing all thirty-one comorbidities) and model *ii* (containing the Australian Injury Comorbidity Index for complications (AICI-comp) with fifteen comorbidities) (Table 5 and S1 Table (SDC3.1)). The CCI (model *iv*) again had the poorest fit.

There was no gain in predictive power by using the lengthy ECM (see Table 5 and S1 Table (SDC3.1)); the AICI-comp with fewer comorbidities was found to yield similar results and had at least a 0.5% advantage in terms of predictive power over the CCI.

The risk-adjusted beta coefficients, incident rate ratios and suggested weights for ICU hours, MV hours and complications are presented in Table 6.

**3.2.3 Complication-type specific comorbidity indices.** Three other comorbidity indices were also derived for the most prevalent complications in the study cohort. These were gastro-intestinal, cardiovascular and metabolic disorders (model results and included comorbidities can be found in S2 and S3 Tables (SDC3.2 & 3.3)). Fifteen comorbidities were found to be associated with the number of complications per the AICI-comp, but only 2–7 conditions were found to show association with the likelihood of specific complications (S2 Table (SDC3.2)). For example, three comorbidities (alcohol dependence, moderate to severe liver disease and valvular disease) were only found to be associated with metabolic disorders and not the other two types of complications. Congestive heart failure was only associated with cardiovascular complications. Pre-existing cardiac arrythmias, chronic pulmonary disorders and uncomplicated hypertension were only associated with cardiovascular complications and

**Table 2. Characteristics of the NSW and WA study populations.**

| | \n | % | At least one comorbidity (%) | Count of comorbidities, mean (95% CI) | n | % | At least one comorbidity (%) | Count of comorbidities, mean (95% CI) |
|---|---|---|---|---|---|---|---|---|
| | Patients admitted[1], n (%) | | | | | | | |
| | July 2012 to June 2014, NSW | | | | July 2012 to June 2014, WA | | | |
| Total patients | 233521 | | | | 84877 | | | |
| **Age group (years)** | | | | | | | | |
| 0–14 years | 31730 | 13.6 | 1.5 | 0.02 (0.01 to 0.02) | 13106 | 15.4 | 1.9 | 0.02 (0.02 to 0.02) |
| 15–24 years | 33888 | 14.5 | 11.0 | 0.13 (0.13 to 0.14) | 13676 | 16.1 | 15.3 | 0.18 (0.17 to 0.19) |
| 25–44 years | 51737 | 22.2 | 13.5 | 0.17 (0.17 to 0.18) | 22617 | 26.6 | 18.6 | 0.23 (0.22 to 0.23) |
| 45–64 years | 44501 | 19.1 | 18.3 | 0.25 (0.24 to 0.26) | 16344 | 19.3 | 21.6 | 0.29 (0.28 to 0.30) |
| 65–84 years | 44948 | 19.2 | 33.2 | 0.51 (0.50 to 0.52) | 12598 | 14.8 | 36.0 | 0.55 (0.54 to 0.57) |
| 85 and over | 26717 | 11.4 | 38.2 | 0.60 (0.59 to 0.62) | 6536 | 7.7 | 41.5 | 0.64 (0.62 to 0.66) |
| **Gender** | | | | | | | | |
| Male[2] | 131598 | 56.4 | 16.2 | 0.23 (0.23 to 0.24) | 49994 | 58.9 | 18.0 | 0.24 (0.24 to 0.25) |
| Female | 101923 | 43.6 | 22.7 | 0.33 (0.32 to 0.33) | 34883 | 41.1 | 24.0 | 0.33 (0.32 to 0.34) |
| **Injury severity[3]** | | | | | | | | |
| Serious injury (ICISS<0.941) | 30265 | 13.0 | 35.5 | 0.57 (0.56 to 0.59) | 9566 | 11.3 | 38.3 | 0.60 (0.58 to 0.62) |
| Other injury (ICISS> = 0.941) | 203256 | 87.0 | 16.6 | 0.23 (0.23 to 0.23) | 75311 | 88.7 | 18.2 | 0.24 (0.23 to 0.24) |
| **Grouped body region** | | | | | | | | |
| Head/face/neck | 48159 | 20.6 | 20.7 | 0.29 (0.28 to 0.29) | 18350 | 21.6 | 23.7 | 0.31 (0.30 to 0.32) |
| Trunk | 29199 | 12.5 | 23.1 | 0.34 (0.33 to 0.35) | 9835 | 11.6 | 23.5 | 0.34 (0.33 to 0.35) |
| Upper extremity | 77389 | 33.1 | 9.4 | 0.12 (0.12 to 0.13) | 27833 | 32.8 | 10.9 | 0.14 (0.13 to 0.14) |
| Lower extremity | 57954 | 24.8 | 19.6 | 0.30 (0.30 to 0.31) | 20678 | 24.4 | 20.9 | 0.31 (0.30 to 0.32) |
| Multiple body regions | 170 | 0.1 | 18.8 | 0.26 (0.17 to 0.35) | 51 | 0.1 | 25.5 | 0.31 (0.15 to 0.47) |
| Unspecified body region | 1247 | 0.5 | 22.3 | 0.34 (0.30 to 0.38) | 376 | 0.4 | 22.1 | 0.30 (0.23 to 0.36) |
| Body region not relevant | 19403 | 8.3 | 45.5 | 0.65 (0.64 to 0.66) | 7754 | 9.1 | 41.6 | 0.56 (0.55 to 0.58) |
| **Grouped injury type (first occurring)** | | | | | | | | |
| Superficial injury | 14875 | 6.4 | 22.6 | 0.33 (0.32 to 0.34) | 4659 | 5.5 | 27.7 | 0.37 (0.35 to 0.39) |
| Open wound | 35691 | 15.3 | 16.9 | 0.23 (0.22 to 0.23) | 12623 | 14.9 | 20.6 | 0.27 (0.26 to 0.28) |
| Fracture | 94258 | 40.4 | 17.3 | 0.26 (0.26 to 0.27) | 30714 | 36.2 | 18.5 | 0.27 (0.26 to 0.27) |
| Dislocation, sprain & strain | 13330 | 5.7 | 6.6 | 0.09 (0.08 to 0.09) | 6755 | 8.0 | 7.5 | 0.09 (0.08 to 0.10) |
| Injury to nerves & spinal cord | 2559 | 1.1 | 8.6 | 0.11 (0.10 to 0.13) | 1147 | 1.4 | 11.1 | 0.14 (0.11 to 0.16) |
| Injury to blood vessels | 1096 | 0.5 | 15.0 | 0.19 (0.16 to 0.22) | 650 | 0.8 | 18.5 | 0.23 (0.19 to 0.28) |
| Injury to muscle & tendon | 10921 | 4.7 | 6.7 | 0.08 (0.08 to 0.09) | 5423 | 6.4 | 10.1 | 0.12 (0.11 to 0.13) |
| Crushing injury | 469 | 0.2 | 3.4 | 0.04 (0.02 to 0.06) | 166 | 0.2 | 4.2 | 0.05 (0.01 to 0.08) |
| Traumatic amputation | 1622 | 0.7 | 6.8 | 0.08 (0.06 to 0.10) | 747 | 0.9 | 7.9 | 0.09 (0.07 to 0.11) |
| Eye injury- excluding foreign body | 919 | 0.4 | 17.3 | 0.22 (0.19 to 0.26) | 308 | 0.4 | 19.5 | 0.23 (0.17 to 0.28) |
| Intracranial injury | 9355 | 4.0 | 28.1 | 0.42 (0.41 to 0.44) | 3385 | 4.0 | 30.6 | 0.44 (0.41 to 0.47) |
| Injury to internal organs | 2198 | 0.9 | 22.5 | 0.32 (0.29 to 0.35) | 1066 | 1.3 | 23.7 | 0.34 (0.29 to 0.38) |
| Foreign body | 3947 | 1.7 | 9.2 | 0.12 (0.10 to 0.13) | 1486 | 1.8 | 9.6 | 0.12 (0.10 to 0.14) |
| Burns | 3244 | 1.4 | 8.8 | 0.12 (0.10 to 0.14) | 1660 | 2.0 | 14.3 | 0.20 (0.17 to 0.22) |
| Other and unspecified injury | 19634 | 8.4 | 19.6 | 0.28 (0.27 to 0.29) | 6334 | 7.5 | 22.8 | 0.30 (0.29 to 0.32) |

(*Continued*)

**Table 2.** (Continued)

| | Patients admitted[1], n (%) | | | | | | | |
|---|---|---|---|---|---|---|---|---|
| | July 2012 to June 2014, NSW | | | | July 2012 to June 2014, WA | | | |
| | n | % | At least one comorbidity (%) | Count of comorbidities, mean (95% CI) | n | % | At least one comorbidity (%) | Count of comorbidities, mean (95% CI) |
| Systemic-poisoning/toxic effects | 17162 | 7.3 | 47.8 | 0.67 (0.66 to 0.68) | 6866 | 8.1 | 43.5 | 0.59 (0.57 to 0.60) |
| Other effects of external cause/complication | 2241 | 1.0 | 28.2 | 0.47 (0.43 to 0.50) | 888 | 1.0 | 26.7 | 0.40 (0.35 to 0.45) |
| **Geographic region** | | | | | | | | |
| Metropolitan Area | 158595 | 67.9 | 18.9 | 0.28 (0.27 to 0.28) | 60151 | 70.9 | 19.7 | 0.27 (0.27 to 0.28) |
| Rural Area | 74918 | 32.1 | 19.2 | 0.27 (0.26 to 0.27) | 24405 | 28.8 | 22.1 | 0.29 (0.28 to 0.30) |
| Unknown | 8 | 0.0 | * | 0.25 (-0.07 to 0.57) | 321 | 0.4 | 34.6 | 0.42 (0.35 to 0.50) |
| **Outcomes** | | | | | | | | |
| Patients requiring an ICU stay | 7000 | 3.0 | 45.1 | 0.78 (0.75 to 0.81) | 774 | 0.9 | 58.0 | 1.02 (0.93 to 1.10) |
| Mean ICU-LOS[7] | 111.71 (98.16 to 125.27) | | | | 85.37 (82.00 to 88.75) | | | |
| Patients requiring MV use | 3300 | 1.4 | 44.5 | 0.73 (0.69 to 0.76) | 1136 | 1.6 | 60.2 | 0.96 (0.89 to 1.02) |
| Mean MV-LOS[9] | 86.07 (73.76 to 98.37) | | | | 75.23 (70.27 to 80.19) | | | |
| Patients with at least one complication | 16277 | 7.0 | 48.8 | 0.85 (0.83 to 0.86) | 7879 | 9.3 | 43.3 | 0.73 (0.71 to 0.75) |
| Complications (median, IQR)[11] | 2 (1 to 3) | | 2 (1 to 3) | 0.86 (0.84 to 0.88) | 2 (1 to 3) | | 2 (1 to 4) | 0.76 (0.74 to 0.78) |
| Complications (mean, 95% CI)[11] | 2.79 (2.75 to 2.83) | | 3.08 (3.02 to 3.14) | | 2.60 (2.54 to 2.66) | | 3.00 (2.91 to 3.10) | |

1. Index injury admissions to all public and private hospitals, limited to residents within the relevant state

2. Intersex/intermediate/unstated sex patient counts less than 5 added to the majority sex group to protect confidentiality

3. Worst injury method-ICD-based Injury Severity Score less than or equal to 0.941 considered as serious injury

4. ICISS–International Classification of Diseases based Injury Severity Scores

5. ICU–Intensive Care Unit

6 LOS–length of stay

7. Includes only patients requiring an ICU stay, excludes children

8. MV–mechanical ventilator

9. Includes only patients requiring an MV use, excludes children

10. Total patients = 161331 for Victoria

11. Includes only patients with at least one complication, excludes children

*Cell count 1–4 suppressed to protect confidentiality

metabolic disorders. This indicates that the associations between comorbidities and complications varies depending on the type of complication and needs due consideration in research and clinical settings.

## 3.3 Comparison of conditions included in new and existing indices

The number of comorbidities associated with in-hospital complications for injury patients in this study were fewer compared to the comorbidities listed in the CCI and ECM. Many of the conditions listed in the CCI and/or ECM, such as HIV/AIDS, drug dependence, blood loss anaemia, malignancies, cerebrovascular disease, deficiency anaemias, diabetes without complications, hemiplegia/paraplegia, complicated hypertension, hypothyroidism, metastatic solid tumors, mild liver disease, myocardial infarction, peptic ulcer disease, pulmonary circulation

**Table 3. Presence of comorbidity and the mean LOS in the ICU and MV, and the mean number of complications (Victoria, ages 15 years and over).**

| Comorbidity | Index admissions (N = 140094), n (%) | ICU stay hours for those[1] using the ICU, mean (CI) | MV use hours for those[2] on the MV, mean (CI) | Complications for those with at least one[3], mean (CI) |
|---|---|---|---|---|
| Human immunodeficiency virus infection and acquired immune deficiency syndrome (HIV/AIDS) | 54 (0.0) | 156.6 (-24.1 to 337.3) | 135.6 (-123.1 to 394.3) | 3.3 (1.5 to 5.0) |
| Alcohol dependence | 6425 (4.6) | 76.8 (68.1 to 85.4) | 52.4 (41.8 to 63.0) | 3.0 (2.9 to 3.2) |
| Drug dependence | 1496 (1.1) | 69.8 (61.4 to 78.3) | 41.8 (33.1 to 50.6) | 2.9 (2.6 to 3.2) |
| Any malignancy | 706 (0.5) | 60.0 (45.6 to 74.4) | 43.1 (22.0 to 64.1) | 3.6 (3.2 to 3.9) |
| Blood loss anaemia | 170 (0.1) | 82.1 (59.6 to 104.7) | 61.8 (17.9 to 105.7) | 3.6 (3.0 to 4.1) |
| Cardiac arrhythmias | 3775 (2.7) | 85.6 (74.9 to 96.3) | 82.2 (57.6 to 106.8) | 3.9 (3.7 to 4.1) |
| Cerebrovascular disease | 602 (0.4) | 123.7 (81.3 to 166.0) | 120.1 (59.8 to 180.4) | 3.6 (3.2 to 4.0) |
| Chronic pulmonary disease | 1302 (0.9) | 106.7 (87.7 to 125.8) | 94.3 (62.0 to 126.5) | 4.4 (4.1 to 4.7) |
| Coagulopathy | 1133 (0.8) | 120.3 (98.5 to 142.1) | 129.3 (96.6 to 162.0) | 3.9 (3.6 to 4.2) |
| Congestive heart failure | 1257 (0.9) | 91.4 (78.5 to 104.3) | 65.5 (45.9 to 85.0) | 4.4 (4.2 to 4.7) |
| Deficiency anaemias | 578 (0.4) | 71.5 (47.9 to 95.1) | 64.5 (9.0 to 120.1) | 3.1 (2.8 to 3.4) |
| Dementia | 2889 (2.1) | 61.4 (49.3 to 73.6) | 76.9 (37.1 to 116.8) | 3.1 (3.0 to 3.3) |
| Depression | 2966 (2.1) | 70.6 (59.9 to 81.3) | 46.0 (36.1 to 56.0) | 2.9 (2.7 to 3.1) |
| Diabetes with chronic complications | 4243 (3.0) | 99.8 (85.6 to 113.9) | 98.7 (73.4 to 124.0) | 3.8 (3.7 to 4.0) |
| Diabetes without complications | 8969 (6.4) | 94.0 (79.3 to 108.7) | 81.7 (62.9 to 100.4) | 3.3 (3.2 to 3.4) |
| Hemiplegia/paraplegia | 538 (0.4) | 105.6 (81.7 to 129.5) | 90.2 (64.7 to 115.8) | 3.6 (3.2 to 4.0) |
| Hypertension complicated | 54 (0.0) | 36.7 (14.0 to 59.4) | * | 4.5 (3.3 to 5.6) |
| Hypertension uncomplicated | 4510 (3.2) | 101.1 (90.8 to 111.4) | 81.2 (67.5 to 94.9) | 4.1 (4.0 to 4.3) |
| Hypothyroidism | 152 (0.1) | 81.2 (47.7 to 114.7) | * | 3.9 (3.2 to 4.6) |
| Metastatic solid tumor | 380 (0.3) | 61.5 (38.9 to 84.1) | 29.4 (-0.6 to 59.4) | 3.3 (2.9 to 3.7) |
| Mild liver disease | 1167 (0.8) | 87.6 (72.5 to 102.6) | 69.2 (51.4 to 87.0) | 3.6 (3.2 to 3.9) |
| Moderate or severe liver disease | 116 (0.1) | 113.4 (74.6 to 152.3) | 107.8 (48.2 to 167.4) | 4.7 (3.8 to 5.7) |
| Myocardial infarction | 248 (0.2) | 104.7 (60.5 to 148.9) | 100.6 (32.4 to 168.8) | 4.1 (3.5 to 4.6) |
| Obesity | 229 (0.2) | 142.4 (92.1 to 192.7) | 174.8 (66.9 to 282.6) | 4.5 (3.7 to 5.3) |
| Peptic ulcer disease | 84 (0.1) | 117.5 (60.0 to 175.1) | 82.6 (33.6 to 131.6) | 4.3 (3.3 to 5.4) |
| Peripheral vascular disease | 702 (0.5) | 124.5 (90.2 to 158.8) | 122.8 (83.1 to 162.5) | 3.7 (3.3 to 4.2) |
| Psychoses | 568 (0.4) | 82.8 (67.4 to 98.2) | 55.2 (34.6 to 75.8) | 3.3 (2.8 to 3.7) |
| Pulmonary circulation disorders | 133 (0.1) | 119.4 (82.6 to 156.1) | 88.0 (30.7 to 145.3) | 4.5 (3.7 to 5.3) |
| Renal disease including renal failure | 3402 (2.4) | 81.2 (72.0 to 90.5) | 61.8 (45.7 to 77.9) | 4.1 (4.0 to 4.3) |
| Rheumatic disease including some other connective tissue disorders | 177 (0.1) | 103.8 (71.5 to 136.0) | 76.8 (14.3 to 139.4) | 3.8 (3.1 to 4.5) |
| Valvular disease | 326 (0.2) | 85.8 (58.1 to 113.6) | 98.0 (49.2 to 146.8) | 4.8 (4.3 to 5.3) |

*Cell count 1–4 suppressed to protect confidentiality

1. n = 5116

2. n = 2413

3. n = 23693

disorders and rheumatic disease, were not associated with ICU hours, MV hours or hospital-acquired complications.

## 3.4 Internal validations

The AICI-icu and AICI-comp were validated in the following subgroups of the study cohort: age group 25–64 years, older adults (> = 65 years), patients with severe injuries (defined using

**Table 4. Presence of comorbidity with CHADx complications (Victoria, ages 15 years and over).**

| Comorbidity | 1: Intra and post procedural | 2: Adverse drug events | 3: Accidental injuries | 4: Specific infections | 5: Cardiovascular | 6: Respiratory | 7: Gastrointestinal | 8: Skin conditions | 9: Genitourinary | 10: Hospital-acquired psychiatric states | 11: Early pregnancy | 12: Labour, delivery & postpartum | 13: Perinatal | 14: Haematological disorders | 15: Metabolic disorders | 16: Nervous system | 17: Other |
|---|---|---|---|---|---|---|---|---|---|---|---|---|---|---|---|---|---|
| HIV/AIDS | * | * | * | * | * | * | 0.1 | 0.1 | * | * | * | * | * | * | * | * | 0.1 |
| Alcohol dependence | 4.7 | 3.6 | 5.2 | 5.9 | 4.1 | 5.5 | 3.7 | 4.4 | 3.5 | 6.6 | 0.0 | 0.0 | 0.0 | 4.5 | 5.8 | 7.3 | 5.0 |
| Drug dependence | 1.5 | 1.2 | 1.3 | 1.1 | 1.4 | 1.6 | 0.9 | 0.8 | 1.0 | 2.4 | 0.0 | 0.0 | 0.0 | 1.0 | 1.4 | 3.4 | 1.3 |
| Any malignancy | 1.2 | 1.4 | 2.0 | 2.3 | 1.4 | 2.2 | 1.7 | 1.5 | 1.5 | 1.8 | 0.0 | 0.0 | 0.0 | 2.0 | 1.7 | 1.9 | 1.7 |
| Blood loss anaemia | 0.5 | 0.4 | 0.4 | 0.4 | 0.5 | 0.4 | 0.5 | 0.6 | 0.6 | 0.6 | 0.0 | 0.0 | 0.0 | * | 0.5 | 0.9 | 0.5 |
| Cardiac arrhythmias | 7.1 | 8.1 | 9.6 | 9.6 | 9.7 | 10.1 | 7.4 | 8.9 | 9.9 | 10.2 | 0.0 | 0.0 | 0.0 | 9.5 | 10.5 | 7.3 | 7.7 |
| Cerebrovascular disease | 1.1 | 0.9 | 2.2 | 2.1 | 1.3 | 1.6 | 1.3 | 1.3 | 1.5 | 1.4 | 0.0 | 0.0 | 0.0 | 1.1 | 1.6 | 2.3 | 1.4 |
| Chronic pulmonary disease | 2.7 | 4.1 | 4.2 | 5.8 | 3.9 | 7.1 | 3.4 | 3.5 | 3.7 | 4.1 | 0.0 | 0.0 | 0.0 | 3.8 | 4.7 | 3.0 | 3.0 |
| Coagulopathy | 2.1 | 2.3 | 2.3 | 3.6 | 2.4 | 2.6 | 1.9 | 2.4 | 2.3 | 2.4 | 0.0 | 0.0 | 0.0 | 2.6 | 2.9 | 3.1 | 2.4 |
| Congestive heart failure | 2.8 | 4.6 | 4.1 | 6.3 | 4.7 | 6.7 | 3.3 | 4.2 | 5.2 | 4.6 | 0.0 | 0.0 | 0.0 | 4.1 | 5.0 | 2.4 | 3.1 |
| Deficiency anaemias | 1.1 | 1.3 | 1.4 | 1.7 | 1.1 | 1.2 | 1.3 | 1.6 | 1.4 | 1.1 | 0.0 | 0.0 | 0.0 | 0.5 | 1.7 | 1.9 | 1.1 |
| Dementia | 4.4 | 4.2 | 8.7 | 6.0 | 6.2 | 6.4 | 4.9 | 6.0 | 8.1 | 7.7 | 0.0 | 0.0 | 0.0 | 8.0 | 7.2 | 3.5 | 4.7 |
| Depression | 1.8 | 2.7 | 2.8 | 3.0 | 2.2 | 2.7 | 2.3 | 1.9 | 2.1 | 3.8 | 0.0 | 0.0 | 0.0 | 1.3 | 3.1 | 4.1 | 3.1 |
| Diabetes with chronic complications | 6.3 | 9.3 | 9.9 | 11.1 | 8.1 | 9.5 | 8.6 | 9.2 | 10.4 | 9.2 | 0.0 | 0.0 | 0.0 | 9.2 | 10.2 | 8.1 | 8.1 |
| Diabetes without complications | 8.8 | 10.6 | 12.1 | 13.6 | 9.7 | 10.9 | 10.7 | 11.6 | 11.3 | 11.0 | 0.0 | 0.0 | 0.0 | 10.6 | 11.9 | 9.8 | 10.3 |
| Hemiplegia/paraplegia | 0.9 | 0.9 | 2.2 | 2.0 | 1.3 | 1.3 | 1.1 | 1.3 | 1.4 | 1.1 | 0.0 | 0.0 | 0.0 | 1.0 | 1.4 | 2.0 | 1.3 |
| Hypertension complicated | 0.2 | 0.2 | * | * | 0.2 | 0.2 | 0.2 | 0.2 | 0.2 | 0.2 | 0.0 | 0.0 | 0.0 | 0.3 | 0.3 | * | 0.2 |
| Hypertension uncomplicated | 8.5 | 12.1 | 14.4 | 14.4 | 13.6 | 12.4 | 11.1 | 12.1 | 14.1 | 13.8 | 0.0 | 0.0 | 0.0 | 13.5 | 15.0 | 13.9 | 11.5 |
| Hypothyroidism | 0.3 | 0.5 | 0.4 | 0.5 | 0.4 | 0.3 | 0.4 | 0.5 | 0.4 | 0.5 | 0.0 | 0.0 | 0.0 | 0.4 | 0.5 | * | 0.4 |
| Metastatic solid tumor | 0.6 | 0.5 | 0.9 | 1.2 | 0.6 | 1.2 | 0.8 | 0.7 | 0.7 | 0.9 | 0.0 | 0.0 | 0.0 | 1.2 | 0.7 | 0.7 | 1.0 |
| Mild liver disease | 1.4 | 1.5 | 1.3 | 1.5 | 1.4 | 1.6 | 1.2 | 1.3 | 1.2 | 1.9 | 0.0 | 0.0 | 0.0 | 1.7 | 1.8 | 2.9 | 1.6 |
| Moderate or severe liver disease | 0.3 | 0.2 | * | 0.7 | 0.4 | 0.4 | 0.3 | 0.5 | 0.3 | 0.5 | 0.0 | 0.0 | 0.0 | 0.6 | 0.6 | 0.8 | 0.3 |
| Myocardial infarction | 0.5 | 0.5 | 1.0 | 0.9 | 1.0 | 0.8 | 0.7 | 0.9 | 0.8 | 0.9 | 0.0 | 0.0 | 0.0 | 0.8 | 0.9 | 0.7 | 0.5 |
| Obesity | 0.7 | 0.7 | 0.4 | 0.8 | 0.5 | 0.8 | 0.5 | 0.7 | 0.7 | 0.6 | 0.0 | 0.0 | 0.0 | 0.4 | 0.6 | 0.8 | 0.6 |
| Peptic ulcer disease | 0.2 | 0.1 | 0.4 | 0.2 | 0.2 | 0.3 | 0.1 | 0.3 | 0.1 | 0.2 | 0.0 | 0.0 | 0.0 | 0.2 | 0.2 | 0.5 | 0.2 |
| Peripheral vascular disease | 1.6 | * | 1.3 | * | 1.1 | 1.1 | 1.0 | 1.6 | 1.0 | 0.9 | 0.0 | 0.0 | 0.0 | 1.2 | 1.1 | * | 1.2 |
| Psychoses | 0.8 | 0.7 | 1.1 | 1.6 | 0.9 | 1.3 | 0.6 | 0.7 | 0.6 | 1.3 | 0.0 | 0.0 | 0.0 | 0.8 | 1.0 | 1.6 | 1.0 |
| Pulmonary circulation disorders | 0.4 | 0.4 | * | 0.7 | 0.5 | 0.7 | 0.4 | 0.5 | 0.3 | 0.0 | 0.0 | 0.0 | 0.0 | 0.4 | 0.5 | 0.8 | 0.4 |
| Renal disease including renal failure | 6.2 | 9.9 | 11.5 | 11.9 | 9.5 | 9.7 | 9.2 | 9.9 | 10.1 | 0.0 | 0.0 | 0.0 | 0.0 | 11.7 | 12.2 | 7.1 | 8.0 |
| Rheumatic disease including some other connective tissue disorders | 0.3 | 0.6 | 0.6 | 0.9 | 0.3 | 0.5 | 0.6 | 0.5 | 0.3 | 0.0 | 0.0 | 0.0 | 0.0 | 0.5 | * | * | 0.4 |
| Valvular disease | 1.0 | 1.1 | 1.6 | 1.3 | 1.3 | 1.6 | 1.0 | 1.5 | 1.1 | 1.3 | 0.0 | 0.0 | 0.0 | 1.4 | 1.5 | 1.0 | 1.2 |

**Table 5. Performance of selected comorbidity measures in assessing the association between comorbidity and selected outcome measures (Victoria).**

| Model | Ln (ICU[1] hours)[2] | | Ln (MV[3] hours)[4] | | Number of complications[5] | |
|---|---|---|---|---|---|---|
| | Adjusted R$^2$ | Model fit AIC | Adjusted R$^2$ | Model fit AIC | Mc. Fadden's Adjusted R$^2$ | Model fit AIC |
| (i) Baseline model | 0.104 | 14591 | 0.145 | 7815 | 0.029 | 106849 |
| (ii) Baseline model + selected comorbidities (individually modelled with binary representation) | 0.121 | 14498 | 0.164 | 7763 | 0.036 | 106142 |
| (iii) Baseline model + selected comorbidities (modelled as a weighted summed score)[6] | 0.121 | 14492 | 0.152 | 7795 | 0.032 | 106584 |
| (iv) Baseline model + comorbidity using CCI weights | 0.108 | 14572 | 0.147 | 7809 | 0.031 | 106639 |
| (v) Baseline model + comorbidity using Quan weights | 0.107 | 14577 | 0.147 | 7810 | 0.030 | 106720 |
| (vi) Baseline model + ECM | 0.130 | 14471 | 0.178 | 7752 | 0.036 | 106103 |

1. ICU–intensive care unit

2. Baseline model includes age, sex, injury severity, injury type and body region; outcome = ICU stay hours (Ln transformed linear model)

3. MV–mechanical ventilator

4. Baseline model includes age, sex and injury severity; outcome = MV hours (Ln transformed linear model)

5. Baseline model includes age, sex, injury type, injury severity and body region; outcome = number of complications for those with at least one complication (negative binomial model)

6. See Methods for weight calculation, excludes weights resulting from an odds <1.2, rounded weights used

See Table 6 for selected comorbidities for each outcome

the ICISS and the worst injury method), and patients with intracranial injuries, hip fractures, blunt and penetrating trauma. The AICI-mv was not validated as the index only included two comorbidities.

**3.4.1 ICU hours.** The AICI-icu was validated on the 25–64 year age group as this group had a relatively high proportion of patients requiring the service [9]. The R$^2$ for this age group and for patients with penetrating trauma was higher than for the full cohort (S4 Table (SDC3.4)) while they were equal or less than the full cohort for the other subgroups. This indicates that the new indices work even better in the 25-64-year age group and patients with penetrating trauma, while it works poorly for hip fracture patients; the latter is expected as these patients are rarely treated in the ICU (only 7.4% of those with hip fractures over the age of 45 years required an ICU stay in Victoria (not shown in Tables).

The performance in terms of the R$^2$ of the AICI-icu was similar to the ECM and CCI in most sub-groups except a few. In the 25-64-year age group, the ECM had the best predictive power, followed by the AICI-icu followed by the CCI, while among hip fracture patients > = 45 years of age, the ECM had the highest predictive power followed by the AICI-icu and CCI.

**3.4.2 Complications.** The AICI-comp was validated on the > = 65-year age group and all other subgroups. For the > = 65-year age group, the R$^2$ was less than the result for the full cohort; even less for severe injuries, intracranial injuries and hip fracture patients; equal for blunt trauma patients; and higher than the full cohort for penetrating trauma patients (S4 Table (SDC3.4)). The R$^2$ of the AICI-comp was higher than that of the ECM and CCI for most of the subgroups except for the > = 65-year age group, intracranial injuries and blunt trauma

**Table 6. Risk adjusted beta coefficients, incident rate ratios, and suggested weights for ICU hours and complications among Victorian hospital-admitted injury patients.**

| | ICU hours | | MV hours | | Number of CHADx complications | | CCI[1] | Updated CCI per Quan et al. (2011) | ECM[2] point scores per van Walraven et al. (2009) |
|---|---|---|---|---|---|---|---|---|---|
| | Beta coefficient (95% CI) | Index weight[3] | Beta coefficient (95% CI) | Index weight[3] | IRR (95% CI) | Index weight[4] | | | |
| **Age groups** | | | | | | | | | |
| 15–24 years | -0.13 (-0.23 to -0.03) | | -0.33 (-0.49 to -0.17) | | 0.69 (0.66 to 0.73) | | | | |
| 25–44 years | -0.03 (-0.11 to 0.06) | | -0.16 (-0.31 to -0.02) | | 0.73 (0.70 to 0.75) | | | | |
| 45–64 years | 0.05 (-0.03 to 0.13) | | 0.12 (-0.03 to 0.27) | | 0.79 (0.77 to 0.82) | | | | |
| 65–84 years | Reference group | | Reference group | | Reference group | | | | |
| 85+ years | -0.26 (-0.36 to -0.16) | | -0.53 (-0.81 to -0.25) | | 1.08 (1.06 to 1.11) | | | | |
| Female gender | -0.03 (-0.08 to 0.03) | | -0.04 (-0.14 to 0.06) | | 0.97 (0.95 to 0.99) | | | | |
| Serious injury | 0.43 (0.35 to 0.51) | | 0.87 (0.77 to 0.97) | | 1.38 (1.34 to 1.41) | | | | |
| **Comorbidity** | | | | | | | | | |
| HIV/AIDS | - | - | - | - | - | - | 6 | 4 | 0 |
| Alcohol dependence | - | - | -0.36 (-0.48 to -0.24)* | - | 1.12 (1.07 to 1.17)* | - | # | # | 0 |
| Drug dependence | - | - | - | - | - | - | # | # | -7 |
| Any malignancy | - | - | - | - | - | - | 2 | 2 | 4 |
| Blood loss anaemia | - | - | - | - | - | - | # | # | -2 |
| Cardiac arrhythmias | - | - | - | - | 1.10 (1.06 to 1.14)* | - | # | # | 5 |
| Cerebrovascular disease | - | - | - | - | - | - | 1 | 0 | ## |
| Chronic pulmonary disease | 0.28 (0.14 to 0.41) | 1 | - | - | 1.27 (1.20 to 1.34) | 1 | 1 | 1 | 3 |
| Coagulopathy | 0.34 (0.19 to 0.50) | 1 | 0.65 (0.38 to 0.93) | 2 | 1.16 (1.09 to 1.24)* | - | # | # | 3 |
| Congestive heart failure | - | - | - | - | 1.17 (1.11 to 1.24)* | - | 1 | 2 | 7 |
| Deficiency anaemias | - | - | - | - | - | - | # | # | -2 |
| Dementia | - | - | - | - | 0.85 (0.81 to 0.88)* | - | 1 | 2 | ## |
| Depression | 0.21 (0.12 to 0.31) | 1 | - | - | 1.15 (1.07 to 1.23)* | - | # | # | -3 |
| Diabetes with chronic complications | - | - | - | - | 1.09 (1.05 to 1.13)* | - | 2 | 1 | 0 |
| Diabetes without complications | - | - | - | - | - | - | 1 | 0 | 0 |
| Hemiplegia/paraplegia | - | - | - | - | - | - | 2 | 2 | 7 |
| Hypertension complicated | - | - | - | - | - | - | # | # | 0 |
| Hypertension uncomplicated | 0.26 (0.17 to 0.36) | 1 | - | - | 1.17 (1.13 to 1.21)* | - | # | # | 0 |
| Hypothyroidism | - | - | - | - | - | - | # | # | 0 |
| Metastatic solid tumor | - | - | - | - | - | - | 6 | 6 | 12 |
| Mild liver disease | - | - | - | - | - | - | 1 | 2 | 11 |

(*Continued*)

**Table 6.** (Continued)

| | ICU hours | | MV hours | | Number of CHADx complications | | CCI[1] | Updated CCI per Quan et al. (2011) | ECM[2] point scores per van Walraven et al. (2009) |
|---|---|---|---|---|---|---|---|---|---|
| | Beta coefficient (95% CI) | Index weight[3] | Beta coefficient (95% CI) | Index weight[3] | IRR (95% CI) | Index weight[4] | | | |
| Moderate or severe liver disease | - | - | - | - | 1.64 (1.38 to 1.94) | 2 | 3 | 4 | 11 |
| Myocardial infarction | - | - | - | - | - | - | 1 | 0 | ## |
| Obesity | 0.62 (0.33 to 0.92) | 2 | - | - | 1.57 (1.37 to 1.80) | 2 | # | # | -4 |
| Peptic ulcer disease | - | - | - | - | - | - | 1 | 0 | 0 |
| Peripheral vascular disease | - | - | - | - | 1.18 (1.08 to 1.30)* | - | 1 | 0 | 2 |
| Psychoses | - | - | - | - | 1.29 (1.16 to 1.43) | 1 | # | # | 0 |
| Pulmonary circulation disorders | - | - | - | - | - | - | # | # | 4 |
| Renal disease including renal failure | - | - | - | - | 1.09 (1.05 to 1.14)* | - | 2 | 1 | 5 |
| Rheumatic disease including some other connective tissue disorders | - | - | - | - | - | - | 1 | 1 | 0 |
| Valvular disease | - | - | - | - | 1.23 (1.12 to 1.36) | 1 | # | # | -1 |

1. CCI–Charlson Comorbidity Index

2. Elixhauser Comorbidity Measure

3. Weight = exp(beta) (see methods for details)

4. Weight = incident rate ratio (IRR) (see methods for details)

- Condition not significantly associated with outcome

# Not included in CCI list

## Not included in ECM list

*Excluded from weighted index as odds <1.2

where it was equal to ECM or CCI. These results indicate that the AICI-comp, ECM nor the CCI is very suitable for serious injury, intracranial injuries or hip fracture patients.

### 3.5 External validations

Characteristics of the two validation cohorts (NSW and WA) can be found in Table 2 and S5 and S6 Tables (SDC3.5 & 3.6). New and existing indices all fared similarly in the validation cohorts, i.e., if the new indices fared poorly, so did the existing indices and vice versa.

**3.5.1 Comparing the performance of the AICI (in Victoria vs NSW and WA).** The AICI-icu's predictive power in the NSW data (7%) was poorer than in the Victorian data (12.1%) while it was much better in the WA data (23%) (S7 Table (SDC3.7)). The AICI-comp's predictive power in the NSW data (2.3%) was poorer than in the Victorian data (3.6%) while it was equal in the WA data (3.7%). Overall, AICIs have validated well in WA but less so in NSW.

**3.5.2 Comparing the performance of the AICIs against the CCI and ECM in NSW and WA.** For ICU hours and complications, the ECM performed best in terms of the $R^2$, followed by the AICI-icu and CCI; but these differences were small.

## 4 Discussion

The association between comorbidities and outcomes varied, depending on the comorbidity, the outcome and how the outcome was measured. Compared to the existing, most widely used index, the CCI, the new (and parsimonious) injury comorbidity indices were able to provide improved predictive power, while compared to the less often used ECM, the new indices performed equally or slightly worse. The new indices however only include comorbidities that are significantly associated with the outcomes, while the CCI and ECM includes comorbidities regardless of their association with the outcome.

### 4.1 Study strengths

This study demonstrated the variation in associations between comorbidity and outcomes, depending on the outcome measure, confirming suggestions from previous studies which recommended study- and outcome-specific comorbidity indices [35–39]. These indices were derived using a population-based database; the indices are current and can be used for general injury patients.

Apart from developing new, outcome-specific comorbidity indices for injury patients, this study also validated and compared some of the most widely used indices such as the CCI, updated CCI and ECM, as well as other methods of measuring comorbidity, such as the presence of at least one comorbidity and the count of comorbidities. In comparing the comorbidities included in each index, it was observed that certain conditions that are listed in the CCI and ECM, such as HIV/AIDS and peptic ulcer disease, were not associated with in-hospital complications in this group of patients. Though the CCI predicts mortality well, very few comorbidities were found to have an actual association with complications outcomes based on the AICIs. It is meaningless to associate comorbidities that have no relevance on the outcomes. The usefulness of indices like these depends on what is being done with them and why they are being employed. However, the CCI and ECM validated with very close predictive powers to the AICIs.

Furthermore, the study also showed that the application of specific weights to comorbidities did not significantly improve the predictive power of regression models above that of the binary representation of the conditions. Similar to the findings by Moor et al. (2008) [40], we found that the weights assigned to comorbidities in the CCI did not correspond to coefficients specific to this study, implying that each study cohort may require an empirical set of weights, if weights are to be used. The AICIs, which are a binary representation, may therefore be more suitable for use, weights are not required. This is in agreement with the conclusions drawn by Moor et al. (2008) [40] and Toson et al. (2015) [21] that binary representation of comorbidities was sufficient for representing the association between comorbidities and injury outcomes such as mortality and resource use.

Since hospital acquired complications may not be specific to injury patients, i.e., complications may be related to treatment and quality of care rather than the primary diagnosis, the AICI-comp could also be tested for use among general hospital-admitted patients.

The parsimonious AICIs are more practical for use in clinical settings and in epidemiology. They use a lesser number of comorbidities than the CCI (in some instances) and ECM in all instances. They are therefore less resource intensive in settings where data has to be collected on comorbidities.

### 4.2 Limitations

**4.2.1 Significance testing.** Significance testing in this study did little to exclude conditions from the models. Due to large sample sizes, most of the significance tests identified significant

associations regardless of the effect size. Instead, to determine which factors were important in the models effect sizes were used in conjunction with the AIC statistic or pseudo $R^2$s to determine the impact of the condition on the overall predictive power of the model.

**4.2.2 Capturing complications.** Hospital acquired complications were captured using the CHADx, which is a coding system used by most Australian hospitals. The CHADx identifies certain diagnosis codes as hospital-acquired complications, with the aid of the main diagnosis codes and a secondary set of codes called the *condition onset flags*. These flags indicate whether the diagnosis was present at admission or occurred during the hospital stay. The number of complications identified in this study is not a perfect estimate of the total, due to: (1) limitations in using hospital administrative databases such as the VAED and (2) limitations of CHADx. Regarding (1): a previous study on the VAED [41] revealed that only 76.2% of admissions were correctly allocated a complication in the 'condition onset' flag, which means that this study could be failing to capture approximately one-quarter of the complications in the Victorian data. The proportion of diagnostic codes supplied with condition-onset flags indicative of a complication varied by state. We found the following proportions of records with a condition on-set flag indicating onset during admission: 18% in Victoria, 8.5% in NSW and 10% in WA. This may have contributed to the comorbidity indices poor validation results in NSW. Regarding (2), the drawback of the CHADx: although it aims to minimise double counting of complications, it has been shown to be less than perfect [42], due to the linear representation of conditions in the diagnosis codes, leaving the possibility for some overestimation of CHADx conditions [42]. Apart from this, some of the complications, although they occur during the hospital stay, may not be related to hospital treatment process, i.e., they could be related to the index condition with a lagged effect. However, in the absence of a more established and robust system for capturing complications (apart from using medical chart review which is not practical in large cohorts), the CHADx is considered sufficient for use with administrative data.

The use of administrative data for hospital-acquired complications surveillance has been criticised as not sufficient, due to the limits imposed by the number of diagnosis codes allowed in a database [43]. However, with 40 or more diagnosis codes in each of the three datasets, this is not considered a limitation in this study.

Hospital-acquired complications could be affected by the hospital facilities, staffing and other variables: information that was not available for inclusion in this study. This information may have improved the baseline models and is recommended for future work to improve the predictive power of the models.

**4.2.3 Capture of comorbidities.** Hospital administrative data has also been criticised for not being able to fully capture all comorbidities for a patient. The main purpose of this type of data is to service administrative and financial planning of hospitals. In this context the coding of comorbidities that may not be actively treated or monitored could be ignored if they were unlikely to incur more resources. Further, the coding does not provide information on the severity of the recorded comorbidities. The reported comorbidities in the administrative data used in this study were only those present at hospital admission; furthermore, conditions were only recorded if they were actively monitored or treated. This situation will, however, improve in the future, as additional codes and requirements for coding comorbidity has been implemented in Australia [44].

Our study did not include lookback periods. However, the inclusion of these is only expected to increase comorbidity capture by about 10% [45]. Lookback periods are impractical in clinical settings, but it may become more increasingly feasible in research settings with data linkage facilities.

The list of comorbidities used in this study was an amalgamation of the CCI and ECM lists, which results in the AICIs, CCI and ECM all performing within a similar predictive power bracket. We carried out a closer investigation of the prevalence rates of ICD-10 codes specific to our study cohort that were not included in the CCI and ECM lists. Codes with a prevalence rate of 1% or more were mainly symptoms such as nausea, vomiting etc and does not amount to chronic conditions and therefore excluded. Lowering the 1% cut-off to 0.5% and investigating those ICD-10 codes for inclusion into the AICIs is recommended for the future.

Model results can also be sometimes misleading if not interpreted with caution. For example, uncomplicated hypertension was associated with all outcomes, over the presence of complicated hypertension, which would not make sense. Uncomplicated hypertension may only get recorded if the patient was in hospital for a long time, and clinical staff become more vigilant in capturing 'everything'. This could result in an over-reporting of this condition, and its presence in the data may display a non-existent association in the model results. This problem can be averted if the severity of the comorbidities could be ascertained.

Given the varying performances of the indices in various subgroups of populations, these indices should be used with care and should not replace clinical judgement.

## 5 Conclusions

The association between in-hospital complications and comorbidities vary with the type of complication and comorbidity. This study derived complication-specific comorbidity indices that are up-to-date, relevant, parsimonious (therefore less resource intensive than existing indices such as the CCI and ECM) and fairly robust. There is room to develop further-improved comorbidity indices for these and other complications, by improving the capture of information regarding both the comorbidities and the complications.

## Supporting information

**S1 Appendix.**
(DOCX)

**S2 Appendix.**
(DOCX)

**S1 Table.**
(DOCX)

**S2 Table.**
(DOCX)

**S3 Table.**
(DOCX)

**S4 Table.**
(DOCX)

**S5 Table.**
(DOCX)

**S6 Table.**
(DOCX)

**S7 Table.**
(DOCX)

## Acknowledgments

The authors would like to thank the Centre for Victorian Data Linkage, the Centre Health Record Linkage in New South Wales and the Western Australian Data Linkage Branch and the custodians of data collections for providing linkage services and the data, and Prof. James Harrison at the National Injury Surveillance Unit for providing the survival risk ratios.

## Author Contributions

**Conceptualization:** Dasamal Tharanga Fernando, Janneke Berecki-Gisolf, Stuart Newstead, Zahid Ansari.

**Formal analysis:** Dasamal Tharanga Fernando, Janneke Berecki-Gisolf.

**Funding acquisition:** Dasamal Tharanga Fernando.

**Methodology:** Dasamal Tharanga Fernando, Janneke Berecki-Gisolf, Stuart Newstead.

**Project administration:** Zahid Ansari.

**Software:** Dasamal Tharanga Fernando.

**Supervision:** Janneke Berecki-Gisolf, Stuart Newstead, Zahid Ansari.

**Validation:** Stuart Newstead.

**Writing – original draft:** Dasamal Tharanga Fernando.

**Writing – review & editing:** Dasamal Tharanga Fernando, Janneke Berecki-Gisolf, Stuart Newstead, Zahid Ansari.

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
