## [Decision Letter · Decision Letter 0]

12 Aug 2020

The Australian Injury Comorbidity Indices (AICIs) to predict in-hospital complications: a population-based data linkage study

PONE-D-20-13280

Dear Dr. Fernando,

We’re pleased to inform you that your manuscript has been judged scientifically suitable for publication and will be formally accepted for publication once it meets all outstanding technical requirements.

Kind regards,

Zsolt J. Balogh, MD, PhD, FRACS

Academic Editor

PLOS ONE

1. Thank you for including your competing interests statement; "All authors declare no support from any organisation for the submitted work. TF has received funding support from the Victorian Injury Surveillance Unit (VISU) at Monash University to pay for two datasets and PhD supervision. No other relationships or activities that could appear to have influenced the submitted work."

Please respond by return email with your amended Competing Interests Statement and we will change the online submission form on your behalf.

2. 

Please include your abstract after the title page.

Reviewers' comments:

Reviewer's Responses to Questions

**Comments to the Author**

1. Is the manuscript technically sound, and do the data support the conclusions?

Reviewer #1: Yes

Reviewer #2: Yes

2. Has the statistical analysis been performed appropriately and rigorously? 

Reviewer #1: I Don't Know

Reviewer #2: Yes

3. Have the authors made all data underlying the findings in their manuscript fully available?

Reviewer #1: Yes

Reviewer #2: Yes

4. Is the manuscript presented in an intelligible fashion and written in standard English?

Reviewer #1: Yes

Reviewer #2: Yes

5. Review Comments to the Author

Reviewer #1: This is an important body of work given the varied professional constructs that can be applied to our understanding of comorbidity. As you note, where the challenges exist often relate to how an indicator is applied (for clinical care, epidemiology, health services planning?). Thank you for your paper.

Reviewer #2: I would like to thank the authors for allowing me to review their manuscript. The quest for parsimony is essential with finite health resources. The ECM, whilst popular in US big data studies, is unwieldly; the CCI perhaps outdated; the quest for simplification in the prediction of adverse outcomes is warranted.

I also thank for the authors for submitting to an open-access journal, and I acknowledge that this is a piece in a higher research degree thesis. The authors have published 3 papers within the last year leading to this manuscript. One of these papers provides a useful review of the predictive abilities of the permutations of the CCI and ECM, another clarified the multiple outcomes such as inpatient mortality, readmission, CHADx complications, cost, LOS, critical care use et cetera in an Australian hospital population of injury. Injury severity was defined by the ICD generated ICISS. The third paper is the generation of a novel score, for which this manuscript tests its predictive ability with the augmentation of select comorbidities and indices.

This paper focuses on the performance of the novel score against ECM and CCI, and it’s validity against other Australian state data sets. It examines the primary outcomes of ICU time, ventilator time and complications (defined by the non-simple CHADx list of ICD-10 codes that had wide variation between states) and develops models through appropriate methods, with specific comorbidities for each outcome. This is a well-balanced paper, with transparent methods, logical presentation of results and a balanced discussion. The supplements contain STATA graphs ad nauseum for the curious. It can be accepted in its current format.

6. PLOS authors have the option to publish the peer review history of their article (what does this mean?). If published, this will include your full peer review and any attached files.

Reviewer #1: No

Reviewer #2: No

---

## [Editor Report · Acceptance letter]

26 Aug 2020

PONE-D-20-13280 

 The Australian Injury Comorbidity Indices (AICIs) to predict in-hospital complications: a population-based data linkage study 

Dear Dr. Fernando:

I'm pleased to inform you that your manuscript has been deemed suitable for publication in PLOS ONE. Congratulations! Your manuscript is now with our production department. 

Kind regards, 

on behalf of

Dr. Zsolt J. Balogh 

Academic Editor

PLOS ONE